# Analysis of Asphalt Mixtures Modified with Steel Slag Surface Texture Using 3D Scanning Technology

**DOI:** 10.3390/ma16083256

**Published:** 2023-04-20

**Authors:** Shuai Zhang, Rongxin Guo, Feng Yan, Ruzhu Dong, Chuiyuan Kong, Junjie Li

**Affiliations:** School of Civil Engineering, Kunming University of Science and Technology, Kunming 650500, China; shuaizh25@163.com (S.Z.); yanfeng@kmust.edu.cn (F.Y.); 18468112733@163.com (R.D.); kongchuiyuan@stu.kust.edu.cn (C.K.); lijunjie1@stu.kust.edu.cn (J.L.)

**Keywords:** steel slag, solid waste, texture structure, 3D scanning technique, height parameters, morphological parameters, resource utilization, skid resistance

## Abstract

This paper investigates the use of steel slag in the place of basalt coarse aggregate in Stone Mastic Asphalt-13 (SMA-13) gradings in the early forming of an experimental pavement and evaluates the test performance of the mixes, combined with 3D scanning techniques to analyse the initial textural structure of the pavement. Laboratory tests were carried out to design the gradation of the two asphalt mixtures and to assess the strength, chipping and cracking resistance of the asphalt mixtures using water immersion Marshall tests, freeze–thaw splitting tests, rutting tests and for comparison with laboratory tests, while surface texture collection and analysis of the height parameters (i.e., Sp, Sv, Sz, Sq, Ssk) and morphological parameters (i.e., Spc) of the pavement were performed to assess the skid resistance of the two asphalt mixtures. Firstly, the results show that a substitution of steel slag for basalt in pavements is a good alternative for efficient resource utilization. Secondly, when steel slag was used in place of basalt coarse aggregate, the water immersion Marshall residual stability improved by approximately 28.8% and the dynamic stability by approximately 15.8%; the friction values decayed at a significantly lower rate, and the MTD did not change significantly. Thirdly, in the early stages of pavement formation, Sp, Sv, Sz, Sq and Spc showed a good linear relationship with BPN values, and these texture parameters can be used as parameters to describe steel slag asphalt pavements. Finally, this study also found that the standard deviation of peak height was higher for steel slag–asphalt mixes than for basalt–asphalt mixes, with little difference in texture depth, while the former formed more peak tips than the latter.

## 1. Introduction

The driving safety of traffic on roads has always been a hot topic in the fields of roads and traffic engineering. With the increasing shortage of stone resources, it has become very important to find new road materials. Steel slag is the slag discharged in the steelmaking process and a by-product of industrial production [1]. The proportion of steel slag produced is about 15–20% of crude steel. By 2021, the total amount of steel slag accumulated in China was about 1 billion tons [2]. Due to the year-round accumulation of steel slag, many regions have become a “slag mountain”, and very few parts can be used [3,4]. In developed countries, such as the United States, Japan, Sweden, Germany and the UK, the utilization rate of steel slag can reach 90%. In contrast, only 30% [5,6,7] is used in China, representing significant room for development. In China, steel slag occupies a lot of precious land resources, and under long-term storage, rainwater leaches in and leaches out heavy metals, contaminating the soil and entering water bodies such as rivers with surface water, which also causes heavy metal pollution of water bodies [8]. Engineering practice in the USA, Japan and Germany [9] has shown that steel slag can be used as an unbonded pavement based on soft subsoils, which not only improves the stiffness of the pavement but, also, the embedded bonding force between the steel slag particles is more conducive to load transfer. China is in a period of rapid construction of high-speed highways. Many stone materials are becoming increasingly limited, and costs are increasing, including basalt, limestone and andesite. If steel slag can be used instead of natural stone resources and applied to asphalt concrete roads, it can provide substantial economic and social benefits.

Steel slag has high strength, and its mechanical properties are much better than rolled crushed stone [10,11]. Steel slag has rich edges and corners, and its surface is rough and porous, providing steel slag with the properties of good particle size distribution and shape [8,12]. Compared with natural aggregates, steel slag has better surface roughness. Compared with some materials, it is hard, wear-resistant and has a good particle shape [13]. Due to the high alkalinity of steel slag, it has strong adhesion to asphalt [14]. After being wrapped in asphalt, it can prevent the expansion of steel slag to a certain extent [15]. Moreover, it has been found that the addition of steel slag in an asphalt mixture would improve mechanical properties [16,17], increase fatigue life [18,19] and improve the noise reduction in an asphalt mixture [20]. It has the potential to replace stone as an asphalt mixture aggregate, which can be used as a high wear resistance pavement material [21].

In the 2010s, it was proven by the United States National Transportation Safety Board (NTSB) and Federal Highway Administration (FHWA) that the main reason for traffic accidents is the lack of anti-sliding friction on the road surface. About 32.3% of traffic accidents occur in wet and slippery conditions on rainy days, of which 70% can be avoided by improving the anti-sliding performance of the road [22,23,24]. In China, about 80% of traffic accidents are related to side slips caused by the insufficient anti-slip ability of the road surface [25]. Research shows that when the anti-sliding level of the road surface is increased by 10%, the traffic accident rate will reduce by nearly 13% [26]. From 2009 to 2010, the Guide for Pavement Friction [27] published by the National Cooperative Highway Research Program provided a systematic summary of research into skid resistance, introducing pavement friction mechanisms, testing and management, proposing a framework for Pavement Friction Management (PFM) and requiring state highway agencies to adopt a strategic and data-driven approach to improve the safety of the road network. Lal found that friction measurement results are affected by micro-texture and macro-texture in 2011 [28,29,30]. Serigos et al. established a prediction model of pavement texture and skid resistance coefficient of the road surface in 2016 [31], which shows a good correlation between pavement texture and British pendulum number. Asphalt pavements have a high anti-sliding capacity at the initial stage, but after a long period of comprehensive action of traffic pressure and the environment, the asphalt thin film gradually falls off, and the aggregate is polished. Tyres cannot produce sufficient adhesion and blocking components during rolling [32,33]. The anti-skid performance of a pavement generally shows an evolution law of first increasing, decreasing and then stabilizing [34,35].

To capture pavement texture, researchers use many non-contact measurement methods, including the digital image method, CT (computed tomography) scanning and laser scanning approaches [36,37,38]. Xiao [39] used digital imaging technology to collect the texture information of asphalt pavements and reconstructed the macroscopic texture shape of the pavement. The CT scanning technology can obtain the internal texture information of the pavement and display the texture data of the pavement. However, CT scanning is expensive, and the detection range is relatively small and limited to laboratory testing [40,41,42]. Qian collected interval asphalt pavement macrotextures by using an EXAscanTM laser scanner and used the pavement texture model ANSYS to evaluate the fatigue performance of the pavement texture by contact calculation with a simplified tire model [43]. At the end of 2015, China built the first field full-scale ring road and used 19 kinds of asphalt pavement structures to study the long-term service performance and evolution law of asphalt pavement structures and materials [44]. Pavement structure includes strong base thin-surface-type semi-rigid base structure, rigid base structure and stress-absorbing layer structure, common forms of asphalt pavement structure, asphalt concrete and semi-rigid base combination form, asphalt concrete and semi-rigid base combination form and full-thickness asphalt pavement structure, etc. The pavement is loaded by means of accelerated loading. The result can be used for research on the equivalent load conversion method, pavement the damage prediction model, the method of structure and material optimal design and maintenance technical standard.

The object of this study was to determine the difference in performance and surface texture of a two-asphalt mixture in the laboratory and on the pavement when steel slag was substituted for basalt. Based on the experimental section of the Kun-Chu highway, the comparative evaluation of the two-asphalt mixture was achieved by combining the British pendulum number test, mean texture depth test, wheel-tracking rutting test, Marshall stability test, freeze–thaw splitting test and swelling characteristic test. Furthermore, the textural structure and texture characteristics of the two-asphalt mixture were evaluated using 3D scanning techniques. From the macroscopic point of view, the texture depth method and the pendulum value method are used; from the microscopic point of view, the initial textures of the two pavements were collected using the 3D scanning technique to observe the differences between the two types of pavements from both perspectives, and it is expected that the pavement textures will need to be collected for the next 1 year to lay the foundation for long-term observation of the decay of the pavement textures.

## 2. Materials and Methods

### 2.1. Steel Slag

Experimental materials of steel slag and basalt were obtained from Kunming, Yunnan, which were measured according to the Chinese standard JTG E42-2005 [45] and assessed for physical properties using the Chinese specification JTG F40-2004 [46]. The results of physical properties are shown in Table 1.

It can be seen from the results in Table 1 that the physical and mechanical properties of the steel slag materials meet the requirements of the quality technology of coarse aggregate for asphalt mixtures for expressways and other levels of class I highway, as well as the requirements of the road surface layer. The crushing value of steel slag is higher than that of basalt, which can show the excellent mechanical properties of steel slag itself. The similar needle and flake content and particle content show that the two aggregates can provide the pavement with sufficient durability.

The factor restricting the use of steel slag in engineering is its expansibility, as the free CaO in steel slag will generate white Ca (OH)_2_ when encountering water, as shown in Figure 1 below. Testing was conducted for free calcium oxide content according to the Chinese standard YBT 4325-2012 [47]. The experimental results are shown in Table 2.

The free CaO content of the steel slag was measured according to the method for the determination of free calcium oxide and the corresponding calculation method for the titration of calcium oxide with EDTA standard solutions at room temperature (25 °C).

The results are shown in Table 2. The average content of free calcium oxide is 1.7%, which meets the requirements of the Chinese standard YBT 4325-2012 [47]. The porous structure of steel slag is conducive to the adsorption of asphalt and aggregate. In order to solve the problem of expansion of steel slag, the engineering solution is to store it outdoors for several months or more than half a year to promote its expansion and damage.

The aggregates used also included fine aggregate limestone as well as filler mineral dust, which were measured according to the Chinese standard JTG E42-2005 [45] and assessed for physical properties using the Chinese specification JTG F40-2004 [46]. The results are shown in Table 3 and Table 4.

### 2.2. Asphalt

The asphalt selected was a styrene–butadiene–styrene (SBS) modified emulsified asphalt, measured according to the Chinese standard JTG E20-2011 [47], meeting the technology requirements of the Chinese specification JTG F40-2004 [46], and the test results are shown in Table 5.

### 2.3. Three-Dimensional Scanner

The scanner equipment used in this article was ZG Scan 717 PLUS, produced by ZG Technology (Wuhan, China). The instrument is shown in Figure 2a.

The handheld scanner is portable and convenient, suitable for various environments and the accuracy meets most requirements. The equipment adopts real-time positioning and mapping technology (i.e., slam technology, simultaneous localization and mapping), does not rely on GNSS positioning such as GPS and can carry out mobile self-positioning and incremental 3D mapping within various unknown environments, such as indoor and outdoor environments. Principle of operation of the scanner: the scanner emits multiple laser lines to the object’s surface, and the binocular camera acquires the laser line image data. After image processing and binocular stereoscopic vision, we obtained the 3D point cloud data of the laser lines’ location on the object surface. As the scanner laser line continuously crosses the object surface, all the laser point clouds form a 3D construction of the object surface. During the operation, we maintained the distance between the scanner and the object at the reference distance, which is the distance between the scanner camera lens and focus point (Figure 2e). This ensured accurate scanning. The depth of field, defined as the clear imaging range of the camera, enabled the scanner to perform 3D scanning at the depth of the field distance (Figure 2f).

The instrument has 28 crossed blue laser lines, which can carry out fast scanning in standard and extensive ranges. The equipment requires point sticking scanning to generate a triangular grid surface automatically. The technical parameters of ZGScan717 PLUS are shown in Table 6.

### 2.4. Gradation Design

SMA-13 gradation is usually used in expressways, and a broken gradation design is used. Because the density of steel slag is higher than that of general aggregate, the mass and volume conversion problems need to be considered in the design process. When basalt and limestone aggregates are used, the density difference between the former and the latter is about 0.2, which does not affect the use of aggregates, and the designed volume ratio is the mass ratio. When steel slag is used as aggregate, the volume ratio is usually used as the initial gradation and then converted to the mass ratio. The design gradation applied to test pavement is presented in Figure 3, with all coarse aggregates of steel slag for type I and all coarse aggregates of basalt for type II. The two grades were laid on a test section. The amount of asphalt used in the two gradations was experimentally determined to be 5.7%. The air voids were measured to be less than 4.0%, meeting the Chinese standard JTG F40-2001.

### 2.5. Methods

#### 2.5.1. The Performance Evaluation of Asphalt Mixture

Firstly, according to the gradation in Figure 3, suitable specimens are made. Further, the rutting test, the water immersion Marshall stability test and the freeze–thaw splitting test were conducted according to the Chinese standard of JTG E20-2011 [48] to evaluate the performance.

The swelling characteristics were tested as follows: the tested specimens were placed in a constant temperature water tank, and the temperature was adjusted to 60 °C and maintained for 7 to 15 days. The height and thickness of the specimens were tested every 3 days; the values were recorded and compared with the initial values to derive the swellability. Test swelling characteristic test was conducted according to the Chinese standard JTG E20-2011 [48] to evaluate the performance.

#### 2.5.2. BPN Tests and MTD Tests in the Laboratory

The method of forming rutting test specimens [48] is as follows: Preheat the rutting apparatus to 100 °C, heat the rutting plate sample mould to the forming temperature, lay the paper sheet, shovel the asphalt mixture from the mixing pot into the mould with the shovel, push the asphalt mixture to the four corners of the mould to fill and compact, place the mould on the rutting apparatus, place the mould on the rutting apparatus, place the paper sheet on the surface of the mould to prevent the asphalt mix from sticking to the rutting apparatus, place the rutting apparatus down and start the rutting apparatus to make 2 laps; then, place the mould in the opposite direction and start the procedure again to make 12 laps. At the end of the rutting, tap the raised part of the surface with a small hammer to keep the specimen flat and remove the paper sheet from the surface. The size of the wheel tracking test mould to be formed in the mean texture depth (MTD) test is 300 mm × 300 mm × 50 mm. The sand laying method was used to carry out the mean texture depth tests on the shaped specimens. The selected part of the experiment is shown in Figure 4. The formed specimens are shown in Figure 5. The zone without wind interference was selected for the tests.

According to the sand patch method, the MTD at the wheel track was measured with 25 mL of standard sand with a particle size between 0.15 and 0.3 mm. The ratio of the sand volume to the circle area is the MTD of the pavement, as shown in Figure 5b, and the test and sampling area of MTD is shown schematically in Figure 4. As shown according to the Chinese specification Asphalt and Asphalt Mixture in Highway Engineering (JTG E20-2011 T0731), 3 specimens were made for each gradation, 3 parallel tests were conducted for each test and the average value of the parallel tests was taken as the final result. During the test, the reduction in sand volume resulted in some deviation in the MTD test results, but the study showed that the deviation was less than 10% and did not affect the comparison [49]. The formula for calculating the texture depth is shown below:(1)$$TD=\frac{1000V}{\pi D^{2}/4}$$
where D is the average diameter of the sand patch in the experiment, mm.

The wheel tracking test mould size required by the British pendulum tests was consistent with the mean texture depth tests, and each British pendulum test was conducted before and after the high-temperature rutting test. Four groups of specimens were made for each gradation. The test part of each testing and sampling area is shown in Figure 4, and the test figure is shown in Figure 6. We took the average value of the results of the two arrow directions as the value of the specimens and took the average value of the results of the specimens in the same group.

The formulas for calculating the pendulum value and attenuation are as follows:(2)$${BPN}_{X1}=({BPN}_{P}+{BPN}_{N})/2$$
(3)$${BPN}_{X}=({BPN}_{X1}+{BPN}_{X2})/2$$
(4)$${BPN}_{XAR}=({BPN}_{Xb}+{BPN}_{Xa})/{BPN}_{Xb}$$
where *X* is the type number; *AR* is the pendulum decay rate, %; *b* represents the value before the rut test; and *a* represents the value after the rut test.

#### 2.5.3. BPN Tests and MTD Tests on Pavement

A 600 m section of the Kun-Chu Expressway was selected for paving, with the test section being a one-way lane. There were 7 groups of measuring points: MP1–Mp6 were type-I mixture with 2 test points each, and MP7 was a type-II mixture with 3 test points. The test locations are shown in Figure 7. In order to meet the road requirements and combine with the surrounding environment, the accuracy of the scanner was set to 0.02 mm~0.03 mm.

The British pendulum number (BPN) tests and mean texture depth (MTD) tests were carried out separately for each of the measuring points shown in Figure 7a.

#### 2.5.4. Three-Dimensional Scanning Tests

The three-dimensional device performed scanning as follows: (a) the test part was placed at the posting point; (b) we started the software and calibrated the device; (c) we performed the scan; and (d) we saved the scanned image. The texture data from the test scans included height parameters (Sp, Sv, Sz, Sq, Ssk) and morphological parameters (Spc). The height parameter Sp refers to the average maximum height of the peaks in the microtexture, Sv refers to the average maximum depth in the microtexture and Sz refers to the average maximum texture depth. A graphical representation of the height parameter is shown in Figure 8. Ssk refers to the parameter of roughness shape (concavity) tendency, and Sq refers to the standard deviation of the height of the points in the area. Spc refers to the average value of the principal curvature of the peak apex points as shown in Figure 9, and the texture parameters are calculated as follows:(5)Sz=Sv+Sp
(6)$$Sq=\sqrt{\frac{1}{A}\iint_{A}^{}Z^{2}\left(x,y\right)dx\;dy}$$
(7)$$Spc=-\frac{1}{2}\frac{1}{n}\sum_{k=1}^{n}\left(\frac{\partial^{2}z(x,y)}{\partial x^{2}}+\frac{\partial^{2}z(x,y)}{\partial y^{2}}\right)$$

The 3D scanner saves the scanned point cloud file, Geomagic Design X software was used to process the scanned point file, the processed file was saved and MountainsMap software was used to analyse the texture parameters and texture values. The 3D texture data were output in visualization. The operation process is as follows: the scanned image was processed using Geomagic Design X to segment and repair it, as shown in Figure 10a. Subsequently, MountainsMap software was utilised to process the 3D texture scan point cloud file of the asphalt pavement and calculate the 3D texture parameters. The software was operated as follows: First, the MountainsMap software was opened and the cloud file was imported to generate the digital model, as illustrated in Figure 10b. Next, the effective area of the scanned sample was extracted using rectangular scale type, as depicted in Figure 10c. Least squares plane levelling was then performed on the effective area to obtain an accurate representation, as shown in Figure 10d. To eliminate any noise and abnormal data, a median of the layered model filtering and noise reduction processing was applied, as demonstrated in Figure 10e. Finally, the 3D visualisation of the scanned sample surface was generated, as presented in Figure 10f.

## 3. Results

### 3.1. The Rutting Test

The results of the dynamic stability from rutting tests for both types of asphalt mixture are shown in Figure 11. The dynamic stability of the steel slag–asphalt mixture is better than that of the basalt–asphalt mixture.

Steel slag itself is a porous structure, and steel slag has strong alkalinity, which can adsorb more asphalt and enhance the high-temperature deformation resistance of the mixture;After production, most steel slag has a large particle size and needs to be broken down. Moreover, the surface of broken steel slag is not smooth, and the rough texture structure can also absorb more asphalt and improve the anti-rutting deformation ability;The needle and flake content of steel slag is higher than that of general aggregate, which indicates that steel slag has rich angularity, has close interlocking ability between aggregates and can also adsorb more asphalt.

### 3.2. The Water Immersion Marshall Stability

Figure 12 shows the immersion stability results for both mixtures. The residual stability of both asphalt mixes is above 80%, and the mixture containing steel slag has significantly better strength before and after immersion. This is due to the chemical composition of steel slag containing CaO, MgO, free CaO and other reactions with water to generate alkali, and its material content is more than 50%; this will cause swelling, but the generated alkali more easily combines with asphalt under the same environment than basalt mixes. It is also more resistant to spalling and more resistant to water damage.

### 3.3. The Freeze–Thaw Splitting Test

Figure 13 shows the results of the freeze–thaw splitting test for the two mixtures. Both asphalt mixtures have a splitting strength ratio higher than 80%, with the mixture containing steel slag having a higher splitting strength.

### 3.4. Swelling Characteristic Test

Table 7 shows the results of the swelling characteristic test. Figure 14 shows a test of the swelling characteristics of the specimens. Both asphalt mixtures swelled to meet the requirements, indicating that the steel slag has a certain degree of stability and can be used in roads, and that the swelling is minimal after the mix is formed and wrapped in asphalt, which obviously controls the swelling rate of the slag.

## 4. Discussion

### 4.1. Surface Mean Texture Depth (MTD)

Table 8 shows the results of the mean texture depth test in the laboratory and on the pavement. In the initial comparison between the two groups, the MTD of the steel slag–asphalt pavement was slightly higher than that of the basalt–asphalt pavement, and the greater the MTD, the better the drainage and skid resistance of the former, and the safer the traffic.

### 4.2. Friction Coefficient (BPN Value)

Table 9 shows the friction test results in the high-temperature test where the tyres were spun 6000 times, and the pendulum decay rate of the steel slag–asphalt mixture was lower than that of the basalt–asphalt mixture, with 4.19%, 4.01%, 4.33% and 3.73% for the first set of trials and an average decay rate of 4.07%; the decay rates for the second set of trials were 5.96%, 5.36%, 5.34% and 6.67%, with an average decay rate of 5.84%. In the early stages of mixing, the former showed better resistance to decay than the latter.

Table 10 shows the British pendulum values for the road surface. The initial pendulum values of steel slag–asphalt mixes are higher than those of basalt–asphalt mixes, both in the laboratory and on pavements, indicating that higher skid resistance can be achieved initially with steel slag materials. Due to the good bonding of the steel slag with asphalt and its good wear resistance properties, it is important to use it in areas with high rainfall or on roads with steep slopes, as well as on motorways.

Due to the excellent bonding ability between steel slag and asphalt and its good wear resistance, it is of great significance to apply it to areas with more rain, highways with steep slopes and expressways;Due to the better combination characteristics of steel slag and asphalt, this mixture can improve the water damage and skid resistance of expressway pavements, prolong the service life of the pavement and reduce the cost and time spent on renovation.

### 4.3. Three-Dimensional Texture Scanning of Pavement

#### 4.3.1. Comparison of Surface Image and Scanned Image

Figure 15 shows the four sets of images selected in the test (actual image on the left, scanned image on the right), and Figure 15d shows the basalt–asphalt mixture for a comparison with the steel slag–asphalt mixtures. It can be seen that the texture picture is consistent with the existing pavement. There are some blank positions of small gaps in the image. These gaps are because the pavement has just been used; the surface is relatively clean and has considerable depth.

In Figure 15, the location of the circular holes is the positioning holes (red circles) required for the measurement process and these require positioning pieces to be determined. It can be seen that the steel slag mix is denser than the basalt mix, with a good skeletal structure and very little large particle to large particle contact, the latter having a smoother surface than the former, with less pronounced surface protrusions. In the process of forming the mix and compacting it, some of the steel slag is crushed, and due to its own porous structure, the greasiness of adsorbed asphalt is reduced. The law of friction shows that the size of the frictional force is proportional to the friction factor, and during the movement of the tyre, the frictional force is divided into sliding friction, where there is relative movement between the tyre and the road surface, and rolling friction, where the tyre itself moves forward, and the surface texture characteristics determine the microscopic texture, determining the size of the adhesion friction between the tyre and the road surface, with the raised part of the surface texture in contact with the tyre rubber, generating adhesive friction; the sharper part generates micro-cutting friction. Figure 16, below, shows macroscopic and microscopic images of simulated tyres on two road surfaces.

As shown in Figure 16, the friction between the tyre and the pavement mainly comes from hysteresis deformation and adhesion in terms of the macrotexture of the surface, and hysteresis deformation friction is the main effect; when the tyre passes over the pavement, it will deform the pavement, and then in order to recover it, it will generate a force perpendicular to the plane outward. Theoretically, the tyre can completely replicate the pavement in the case of a finer texture or greater elasticity, but in practice, the tyre will only come into contact with the highest point of the road surface, as shown in Figure 17, and the crest of the texture is more susceptible to the effect of traffic polishing than the valley, which generates friction, which has a great impact on the skid resistance of the road surface. In the case of rain, the rougher the texture of the road surface, the greater the deformation of the tyre material will be, and the rainwater will accumulate in the texture, increasing the hysteresis coefficient of friction. Reducing the contact between the accumulated water and tyres increases the anti-skid ability of the road surface and reduces the occurrence of accidents. Since the microtexture of the pavement is too fine compared with the macrotexture structure, the microtexture structure mainly provides friction for low-speed vehicles, while the current measurement of highway pavement texture mainly considers the macrotexture.

Van der Waals forces influence the skid resistance of the pavement, adhesion forces, the elastic deformation of tyres and micro-cutting forces acting on micro convex bodies [50]. Steel slag–asphalt mixes have more bumps, which are prone to stress concentration and more pronounced micro-cutting action, which provides friction. It is known that the hindrance force accounts for about 10% of the tyre–road friction [51], and the cutting effect of micro-convexity accounts for about 90% of the overall friction [52,53].Vehicles traveling on highways at speeds greater than 60 km/h are greatly influenced by the thickness of the water film, and a water film of about 0.025 mm can reduce the friction factor by roughly 20% to 30%, a situation that occurs when rainfall is greater than 0.25 mm [54], which can cause the water drifting phenomenon to occur.The use of industrial wastes on roads greatly reduces the exploitation of scarce resources, reduces the cost of materials and meets the national requirements for environmental protection, enabling the conversion of waste into valuable materials.

#### 4.3.2. Three-Dimensional Texture Analysis

Figure 17 shows the surface 3D textures of the two types of mixes after the noise reduction treatment. Figure 18a–c display the steel slag–asphalt mixes and Figure 18d,e display the basalt–asphalt mixes. The size of the intercepted scanned images is 100 mm × 50 mm.

For the steel slag–asphalt mixture and basalt–asphalt mixture, it can be seen from Figure 19a that the Sp value of the former is higher than that of the latter, indicating that the former has a higher peak than the latter, and the surface texture is more prominent. There is no difference between the two Sv values, and the difference in depth is not significant, indicating that there is no difference between the two texture depths in the initial stage of forming. Sz describes the distance from the road surface to the bottom of the valley, which is the total value of the structural layer of the road section. The middle layer plus the surface layer is expected to be laid at a height of 8.5, and the value of Sz is also basically around 8.5, indicating that the test section meets the laying requirements. From Figure 19b, it can be seen that the standard deviation of the former height is higher than that of the latter, indicating that the peak variability in the former is greater, the peak height of the mix surface is staggered, the surface roughness is higher and the slip resistance of the former is higher than that of the latter. From Figure 19c, it can be seen that both Ssk values are negative, indicating that both height distributions are above the average surface and there is no greater variability. From Figure 19d, it can be seen that the Spc values of the former are both higher than those of the latter, indicating that the former has a higher principal curvature at each peak point, and the area in contact with the object is smaller, with sharper peaks and greater skid resistance. In summary, the textural structure of the two pavements is somewhat different, with the steel slag–asphalt mix having better skid resistance than the basalt–asphalt mix in the early stages of pavement formation.

The correlation between the texture characteristic parameters of each measuring point (Mp1–Mp6, excluding Mp7) and British pendulum during the initial pavement of the steel slag mixture was analysed and fitted using a linear equation as follows:(8)$$BPN=aT_{x}+b$$
where *a* and *b* are regression coefficients, and *T_x_* is 3D texture parameters. The fitting results are shown in Figure 20, and the fitted parameters are shown in Table 11.

From the fitting results, it can be seen that Sp, Sz, Sv, Sq and Spc generally correlated well with British pendulum, and their coefficient of determination (R^2^) reached 0.85, 0.87, 0.85, 0.97 and 0.81, respectively. They were all >0.8, indicating extremely strong correlation. It is suitable to use Sp, Sz, Sv, Sq and Spc to describe the relationship of the 3D texture and skid resistance.

## 5. Conclusions

In this study, the use of steel slag as a substitute for basalt in asphalt mixes was investigated and evaluated for mix performance through laboratory experiments. The two types of mixtures were also laid on the experimental pavement and combined with 3D scanning technology to analyse the pavement texture structure.

The free CaO content of the steel slag used in this study meets national standards and can be used for road surfaces. The performance enhancement in the mix by steel slag was superior to that of the conventional basalt–asphalt mix, with a reduction in friction value decay from 5.7% to 4% before and after high-temperature tests with steel slag replacing the conventional basalt aggregate, significantly improving the skid resistance of the pavement.The steel slag asphalt mixture has better performance than the basalt mixture in all properties, with the former having 14% higher dynamic stability than the latter; the former having 27% higher residual stability than the latter; the former having 10.5% higher freeze–thaw resistance than the latter; and the former having 0.16% higher swelling characteristics than the latter, with almost no effect.In addition to comparing the texture values of the two mixes, the test results show that the texture depths of the two mixes are basically the same, and both textures show a tendency to protrude, although the steel slag–asphalt mix pavement has a sharper and rougher surface texture than the basalt–asphalt mix, with better skid resistance, and the maximum height Sz value can reflect the thickness of the pavement structure layer and verify whether the pavement matches the paving requirements.This paper combines 3D scanning technology to investigate the initial skid resistance of the pavement from the analysis of the macro road performance of the mixture to the analysis of the microtexture data. In the early stages of pavement formation, there is a good correlation between pavement texture parameters and BPN values.The swelling characteristics of steel slag are controlled to ensure that it does not crack when used in roads, and the use of the steel slag asphalt mixture for paving and pavement repair has high economic benefits and will also gradually reduce the demand for natural aggregate extraction. Analysing the textural structure of the surface of steel slag asphalt mixes and establishing the relationship between texture and skid resistance are a guide to the use of steel slag on pavements. With the long-term use of the experimental pavement, the collection of texture values at the same locations will be considered in future tests to investigate the skid decay of the experimental pavement and to establish a skid decay model for both mixtures.

## Figures and Tables

**Figure 1 materials-16-03256-f001:**
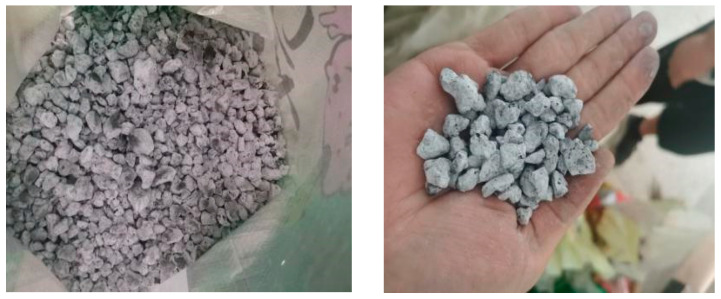
Steel slag alkalization.

**Figure 2 materials-16-03256-f002:**
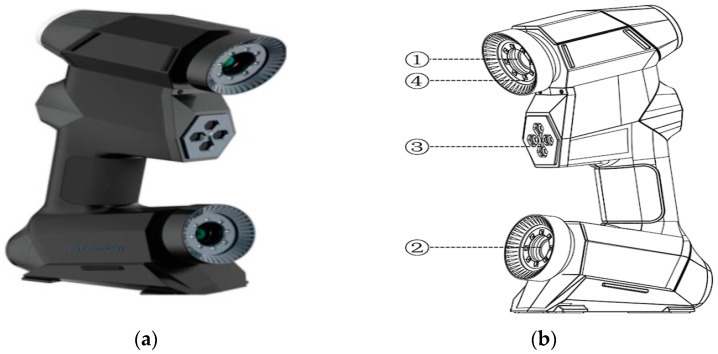
(**a**) Three-dimensional topography instrument; (**b**) ① camera A; ② camera B; ③ transmitter optical; ④ LED light supplement lamp; (**c**) ⑤ ring coloured indicator; ⑥ function buttons; ⑦ power plug; ⑧ socket transfer data; (**d**) pavement scanning; (**e**) scanning distance; (**f**) depth of field distance.

**Figure 3 materials-16-03256-f003:**
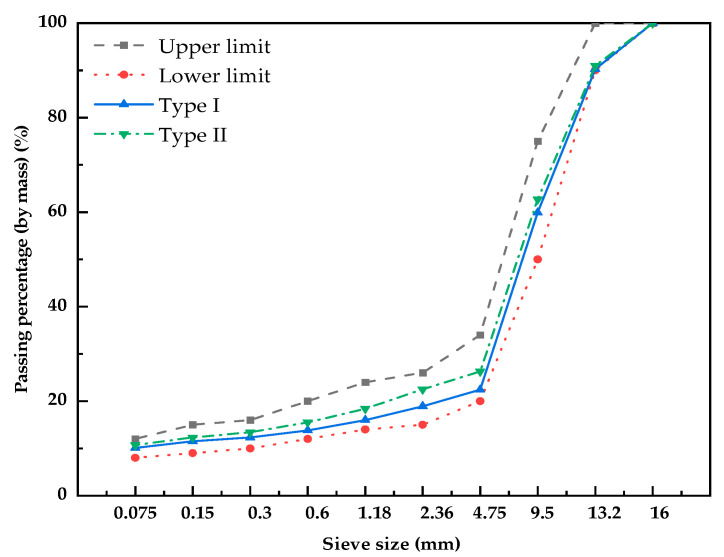
Gradation of two types.

**Figure 4 materials-16-03256-f004:**
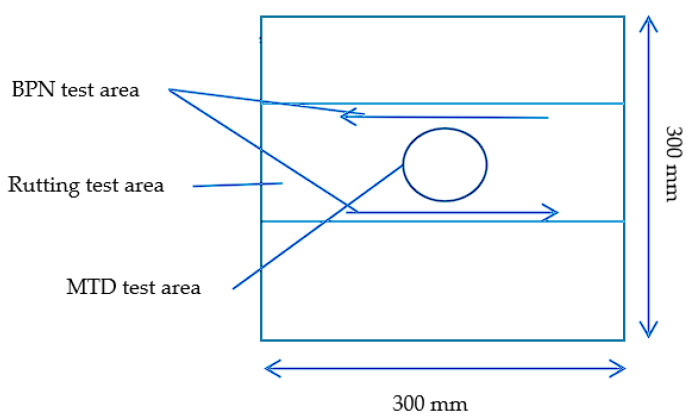
Experimental selection section.

**Figure 5 materials-16-03256-f005:**
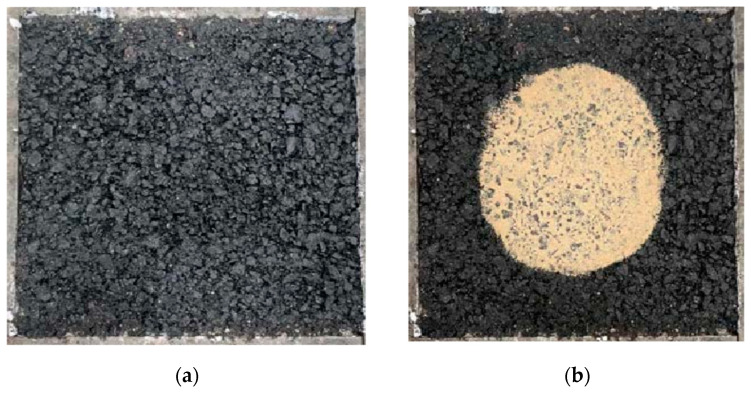
Surface topography of surface layers and mean texture depth test: (**a**) rutting test specimens; (**b**) mean texture depth (MTD) test area.

**Figure 6 materials-16-03256-f006:**
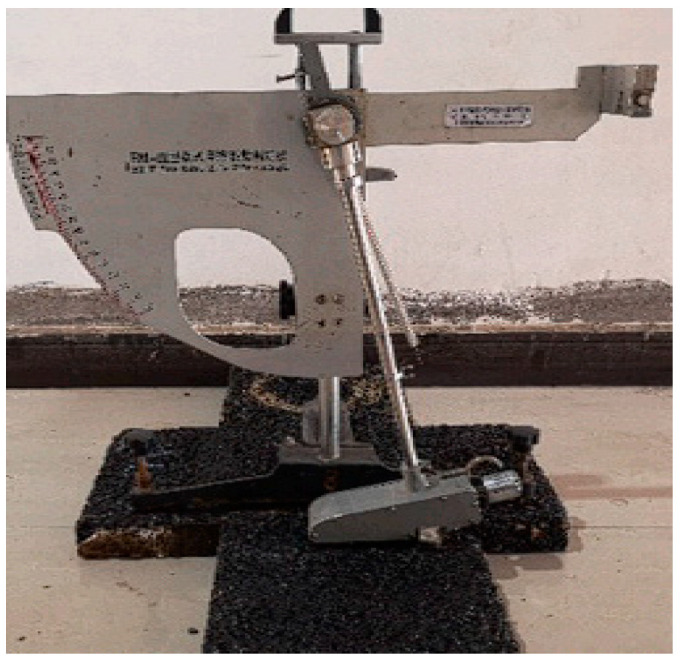
British pendulum test.

**Figure 7 materials-16-03256-f007:**
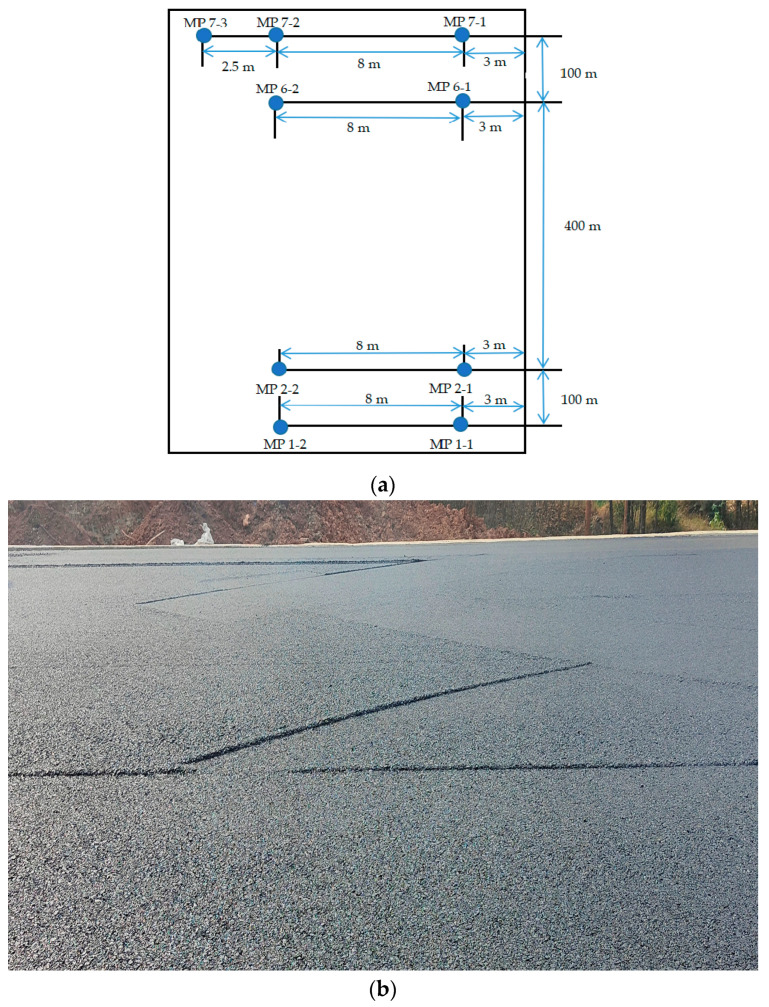
Location of road test points: (**a**) road test area; (**b**) actual road surface.

**Figure 8 materials-16-03256-f008:**
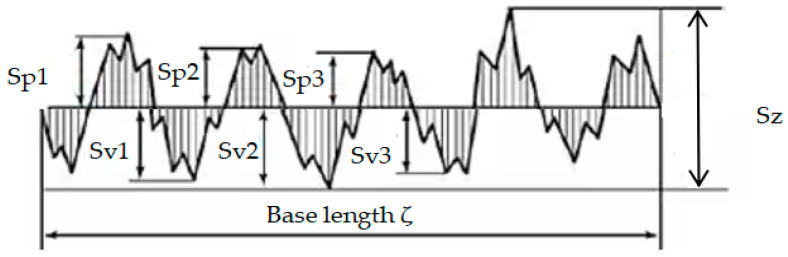
Height parameter graph.

**Figure 9 materials-16-03256-f009:**
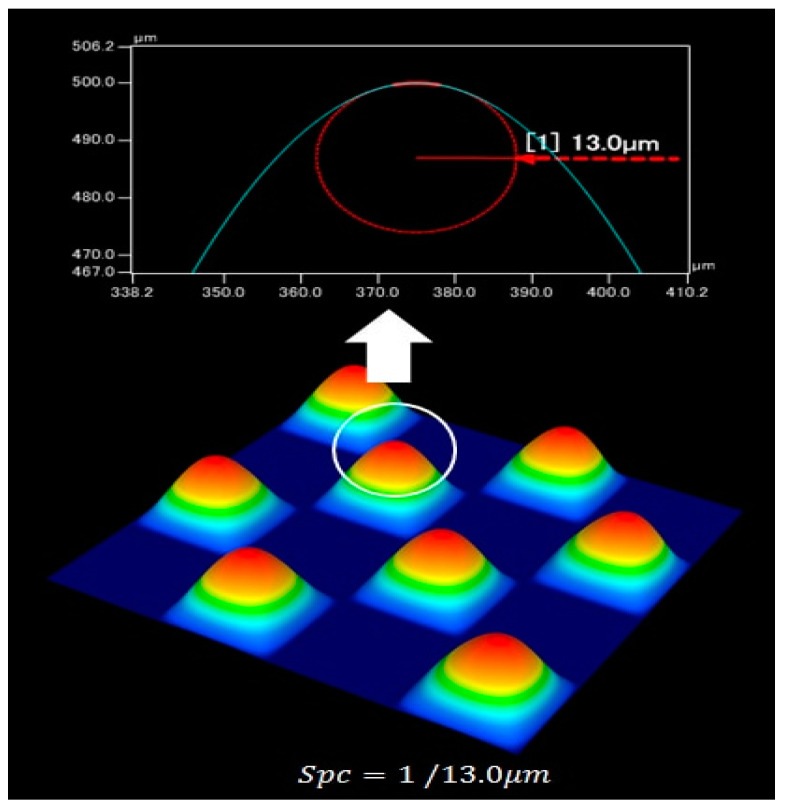
Arithmetic mean curvature of the crest.

**Figure 10 materials-16-03256-f010:**
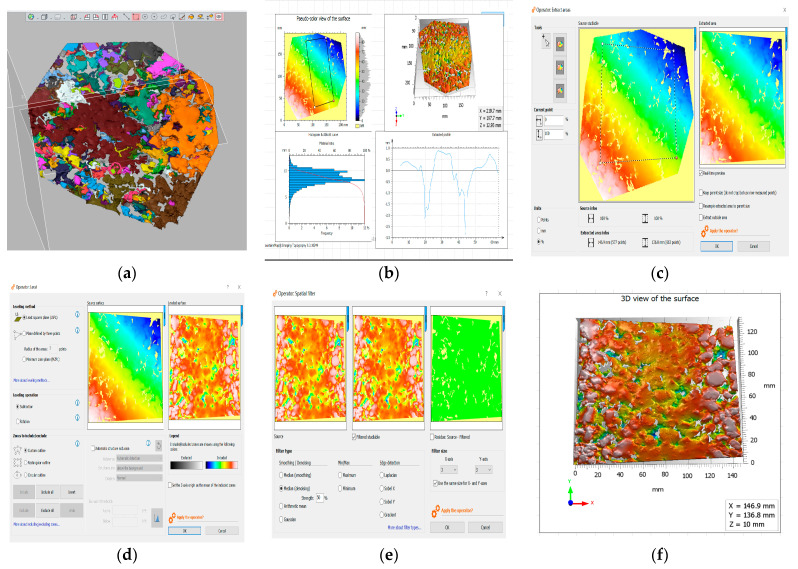
Calculation process for asphalt pavement surface 3D texture: (**a**) processing image; (**b**) importing cloud files; (**c**) extracting valid areas; (**d**) applying least squares plane levelling; (**e**) performing median-filtering noise reduction; (**f**) generating 3D visualised views.

**Figure 11 materials-16-03256-f011:**
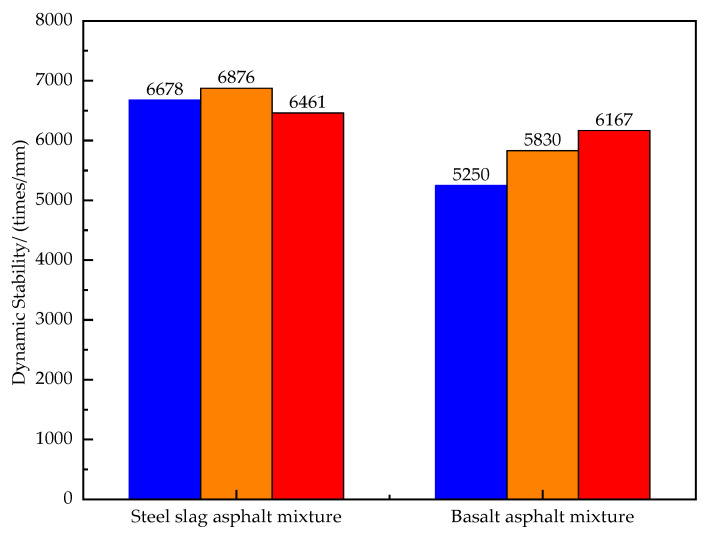
Dynamic stability of two types of asphalt mixture.

**Figure 12 materials-16-03256-f012:**
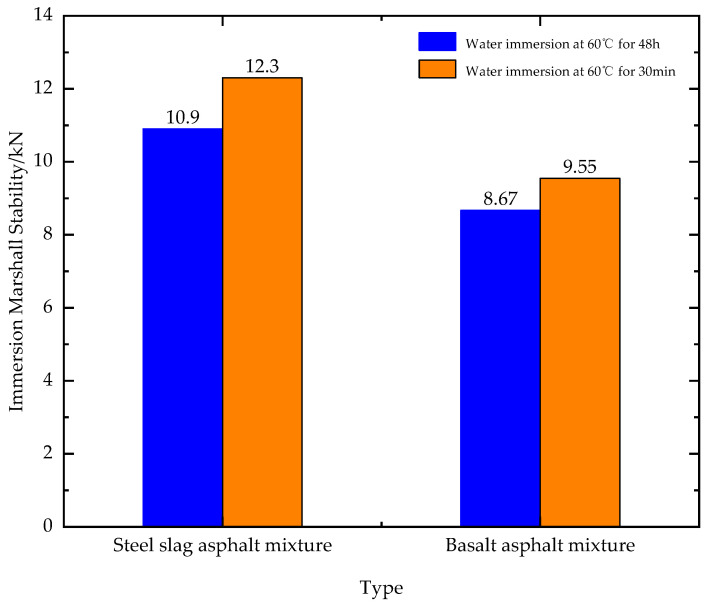
Immersion Marshall stability of two types of asphalt mixture.

**Figure 13 materials-16-03256-f013:**
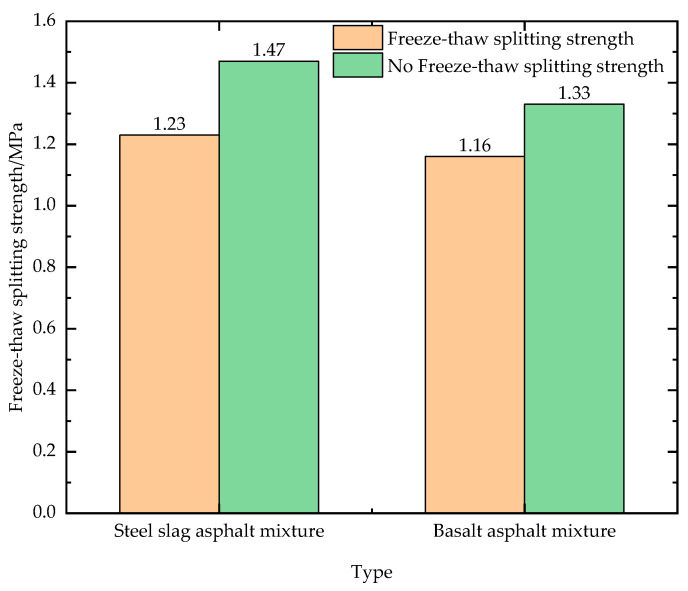
Freeze–thaw splitting test of mixture.

**Figure 14 materials-16-03256-f014:**
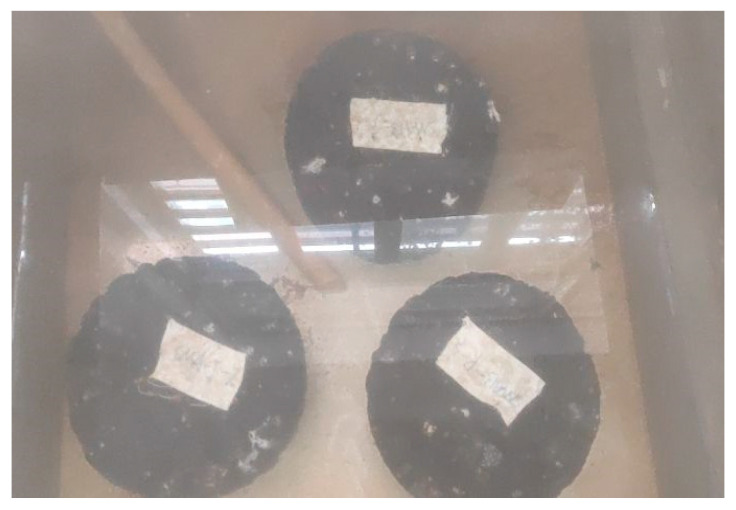
Swelling characteristic test.

**Figure 15 materials-16-03256-f015:**
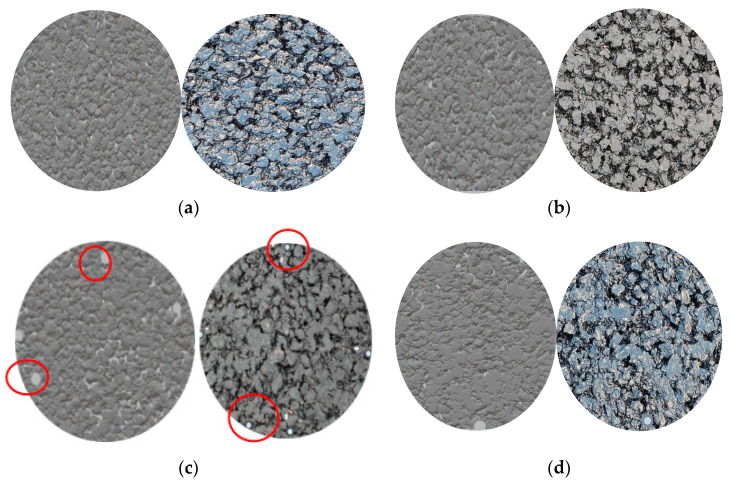
Scanned and actual images of the two mixtures (The red circles show the positioning holes.). (**a**–**c**) Steel slag–asphalt mixture; (**d**) basalt–asphalt mixture.

**Figure 16 materials-16-03256-f016:**
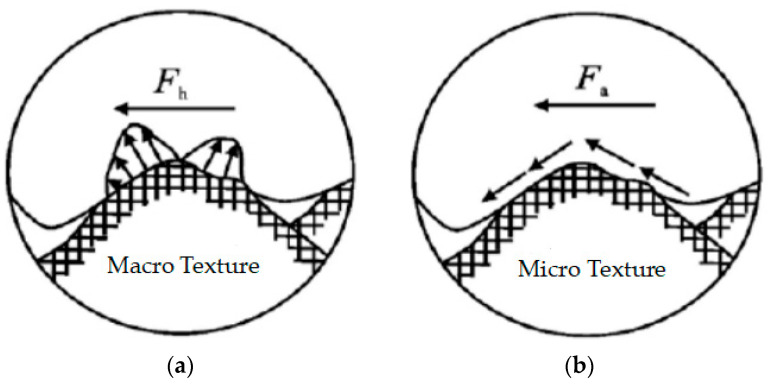
Pictures of tyre textures on two road surfaces. (**a**) Macrotexture; (**b**) microtexture.

**Figure 17 materials-16-03256-f017:**
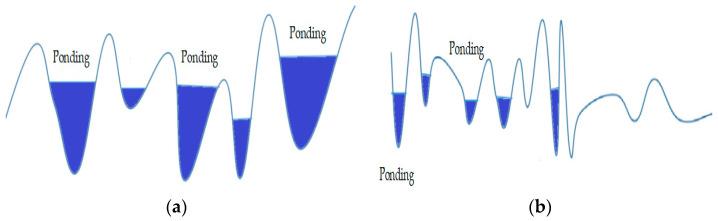
Two types of mixes with water logging methods. (**a**) Basalt–asphalt mixes textured with ponding water; (**b**) steel slag–asphalt mixes textured with ponding water.

**Figure 18 materials-16-03256-f018:**
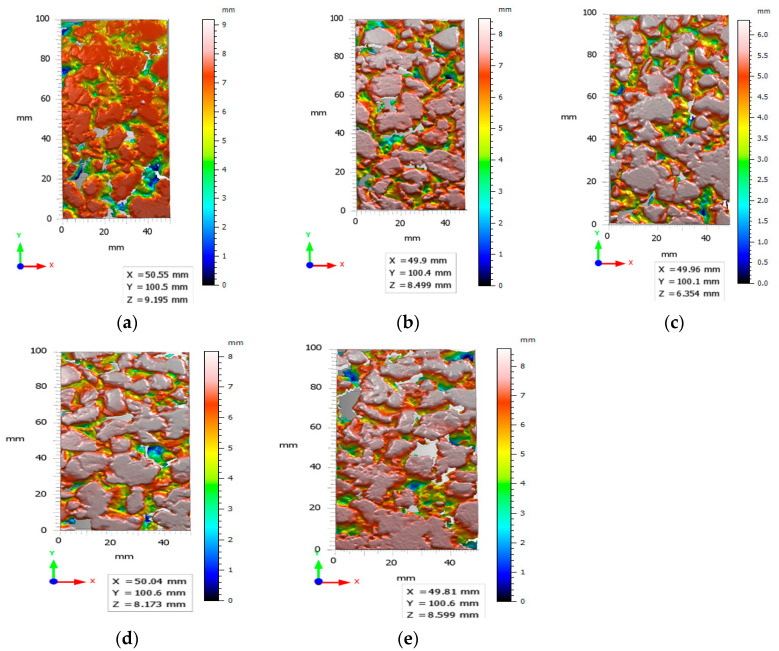
Surface 3D textures of several sets of measurement points. (**a**) Mp 1-1; (**b**) Mp 4-1; (**c**) Mp 6-1; (**d**) Mp 7-1; (**e**) Mp 7-2.

**Figure 19 materials-16-03256-f019:**
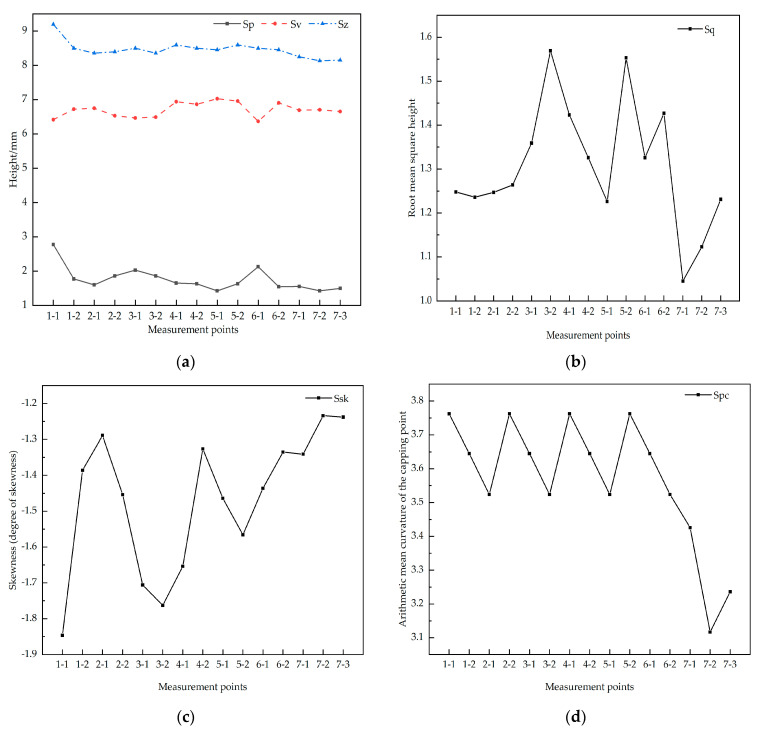
Texture values for different measurement points. (**a**) Sp, Sv, Sz for different measurement points; (**b**) Sq for different measurement points; (**c**) Ssk for different measurement points; (**d**) Spc for different measurement points.

**Figure 20 materials-16-03256-f020:**
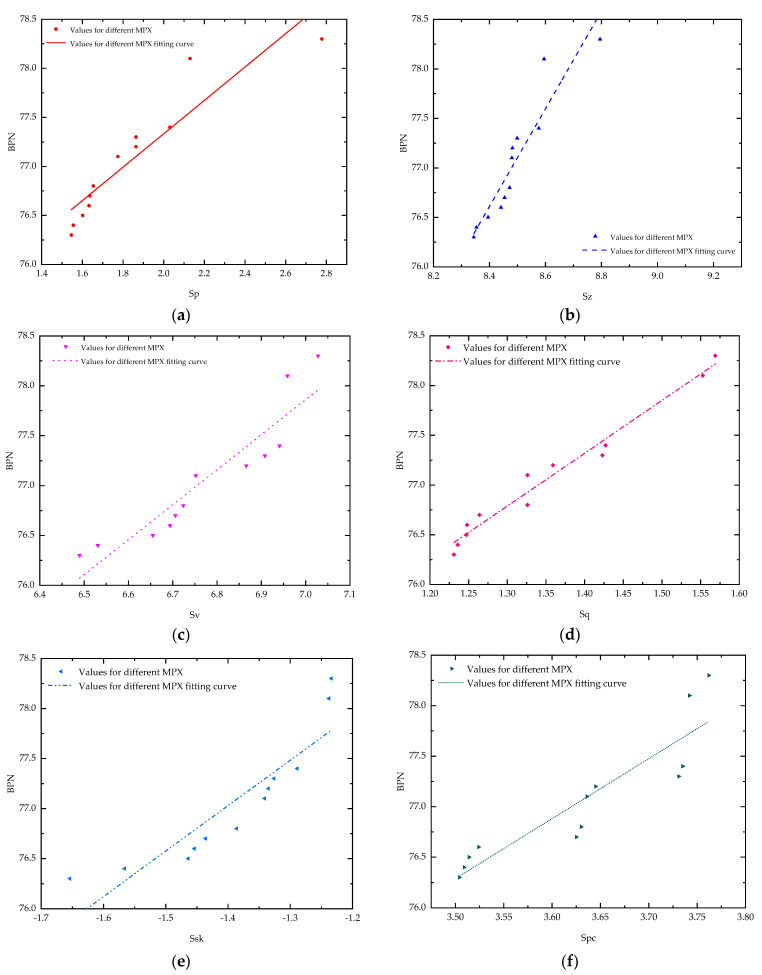
Regression analysis of initial 3D texture parameters of pavement and British pendulum: (**a**) Sp; (**b**) Sz; (**c**) Sv; (**d**) Sq; (**e**) Ssk; (**f**) Spc.

**Table 1 materials-16-03256-t001:** Technical indices of the basalt and steel slag.

Index		Steel Slag	Basalt	Spec Requirements [46]
Crushing value		12.8	13.8	≤28
Los Angeles wear value (%)		10.2	17.9	≤30
Apparent relative density	Particle size (12–17 mm)	3.553	2.916	≥2.50
	Particle size (7–12 mm)	3.604	2.928	
Water absorption	Particle size (12–17 mm)	0.873	1.209	≤30
	Particle size (7–12 mm)	0.962	2.611	
Percentage of flat-elongated particles (%)		8.9	6.7	≤18
Washing particle content (<0.075 mm)		0.09	0.33	≤18
Content of soft rock		1.7	0.5	≤5
Adhesion grade		5	5	≥4

**Table 2 materials-16-03256-t002:** Free CaO content.

Aggregate Batches	Free Calcium (%)	Ca (OH)_2_/%	Free-CaO/%
1st batch	1.64	0.0016	1.640
2nd batch	2.03	0.0075	2.020
3rd batch	0.67	0.0047	0.660
4th batch	1.32	0.023	1.300
5th batch	1.45	0.0016	1.450
6th batch	1.46	0.0043	1.450
7th batch	1.96	0.0046	1.950
8th batch	2.11	0.0036	2.100
9th batch	1.71	0.0066	1.701

**Table 3 materials-16-03256-t003:** Technical indices of the limestone.

Index		Results	Spec Requirements [46]
Apparent relative density	>2.36 mm	2.806	≥2.50
Water absorption (%)		1.344	/
Gross volume relative density		2.704	/
Ruggedness (%)		4	≤12
Methylene blue number (g/kg)		1	≤1.4
Sand equivalent (%)		74	≥60
Angularity (flowing time (s))		35.7	≥30

**Table 4 materials-16-03256-t004:** Technical indices of the filler.

Index		Results	Spec Requirements [46]
Apparent relative density (t/m^3^)		2.816	≥2.50
Water content (%)		0.4	≤1
Appearance		No agglomeration	No agglomeration
Hydrophilic coefficient		0.7	<1
Index of plasticity		2.7	<4
Heating stability		No change in colour	Physical record
Particle size range (%)	<0.6 mm	100	100
	<0.15 mm	99.7	90~100
	<0.075 mm	96.5	75~100

**Table 5 materials-16-03256-t005:** Technical indices of the asphalt.

Index		Results	Spec Requirements [46]
Thixotropic index (25 °C, 5 s,100 g)		49	40~60
Penetration index PI		0.309	≥0
Ductility (5 cm/min, 5 °C) (cm)		29	≥20
Softening degree (global method) (°C)		75	≥60
135 °C kinematic viscosity (Pa·s)		1.55	≤3
Flash point (°C)		295	≥230
Density (15 °C) (g/cm^3^)		1.03	Physical record
25 °C elastic recovery (%)		83.1	≥75
Storage stability segregation, 48 h softening point difference (°C)		2	≤2.5
Thin-film heating test 163 °C, 5 h	Quality loss	−0.0304	±1.0
	Residual penetration ratio (25) (%)	76.4	≥65
	Residual ductility (5 cm/min, 5 °C) (cm)	59	≥15

**Table 6 materials-16-03256-t006:** Technical parameters of ZGScan717 PLUS.

Equipment Composition	Technical Parameter	Numerical
Scanning head	Measurement rate (standard mode)	1,350,000 times/s
	Measurement rate (fine mode)	450,000 times/s
	Scan area	Max 600 × 550 mm
	Resolution	≥0.01
	Accuracy (standard mode)	Up to 0.02 mm
	Accuracy (fine mode)	Up to 0.01 mm
	Volumetric accuracy (standard mode)	0.02 + 0.035 mm/m
	Volumetric accuracy (fine mode)	0.02 + 0.015 mm/m
	Datum distance (standard mode)	300 mm
	Datum distance (fine mode)	150 mm
	Depth of field (standard mode)	450 mm
	Depth of field (fine mode)	150 mm
	Maximum depth of field	550 mm
Hardware specifications	Operating temperature range	−20~40 °C
	Working humidity range	10–90%

**Table 7 materials-16-03256-t007:** Expansion test of asphalt mixture.

Gradation Type	The Average Value of Swelling	Spec Requirements
Steel-slag SMA-13	0.59%	≤1.0%
Basalt SMA-13	0.43%	

**Table 8 materials-16-03256-t008:** The MTD value of both pavements.

Gradation Type		Laboratory/mm	Pavement/mm
Steel-slag SMA-13	Group 1	0.67	0.64
	Group 2	0.65	0.63
	Group 3	0.64	0.63
Basalt SMA-13	Group 1	0.63	0.63
	Group 2	0.60	0.61
	Group 3	0.61	0.60
Regulation		≥0.55	

**Table 9 materials-16-03256-t009:** British pendulum values in the laboratory.

Gradation Type		Before the Rutting Test	After Rutting Test	Attenuation Rate/%
Steel-slag SMA-13	Group 1	76.4	73.2	4.19
	Group 2	77.3	74.2	4.01
	Group 3	76.3	73	4.33
	Group 4	75.1	72.3	3.73
Basalt SMA-13	Group 1	78.6	74.7	5.96
	Group 2	77	73.1	5.36
	Group 3	76	72.7	5.34
	Group 4	75	70	6.67

**Table 10 materials-16-03256-t010:** British pendulum values for the road surface.

Gradation Type		BPN Value
Steel-slag SMA-13	MP 1	78.3
	MP 2	77.4
	MP 3	76.6
	MP 4	76.3
	MP 5	76.8
	MP 6	77.3
Basalt SMA-13	MP 7	76.1

**Table 11 materials-16-03256-t011:** Fitting parameters of initial 3D texture parameters of pavement and British pendulum.

Parameter	Sp	Sz	Sv	Sq	Ssk	Spc
a	73.925	35.19	53.307	69.892	83.382	55.503
b	1.704	4.931	3.508	5.306	4.538	5.939
R^2^	0.85	0.87	0.85	0.97	0.74	0.81

## Data Availability

Data will be made available on request.

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
