# Peer review of "Analysis of Asphalt Mixtures Modified with Steel Slag Surface Texture Using 3D Scanning Technology"

_materials, 2023, doi:10.3390/ma16083256_

Round 1
Reviewer 1 Report
The article analyzes the texture and friction of a bituminous mixture produced, partly with steel slag aggregate, comparing the results with those of a bituminous mixture with basaltic aggregate.
Although the article contributes to promoting the use of steel slag in China, it lacks a remarkable level of innovation. Despite this, using a 3D scanning technique to evaluate the surface texture stands out.
I think the article deserves to be published, despite having some weaknesses that can be improved before it is accepted for publication.
Firstly, the article does not have very recent bibliographical references, showing some lack of novelty. Furthermore, the knowledge gap that the authors fill with the paper is not clear.
Regarding the terminology used, unusual terms are presented. I suggest that the authors improve these aspects. Consider revising:
- "rut plate" to "Wheel tracking test mould";
- "construction depth" to "texture depth";
- "flattened sand" to "sand patch";
- "rut experiment" to "wheel tracking test";
- "height of the texture" to "texture depth";
- "structural depth test"?
Since tests have different designations and standards in other areas, authors should include the standards used in all tests whose results they present. In addition, test conditions were not mentioned in several cases. Examples: How were the slabs shown in Fig 4 compacted? What test temperature was used for the splitting tests (Fig 11)? What is the water temperature during the immersion period (Fig 12)?
Are the units of the results in Table 10 mm?
Table 11 presents the "after-rutting test" results. A "rutting test" is a wheel tracking test to assess the resistance of mixtures to permanent deformation. The purpose is not clear in this context.
Author Response
Please see the attachment.
I have uploaded the revised PDF file in the attachment. If there is anything in the responses that you do not understand, you can view it in the latest PDF and WORD.

Reviewer 2 Report
Please find below a list with my comments for this interesting study.
1_ First of all, define the type of the paper in the line before the title (i.e., I guess this an article!!). Add full line numbering within the text and define the lines of any kind of modification in your replies after this revision in order to facilitate any re-review process.
2_ Improve the use of English. Some sentences are very lengthy, e.g., the 2nd sentence of the abstract (In-lab tests… the two mixes). In addition, many phrases are not scientifically correct, e.g., “hot topic”, “room for development” in the introduction, 1st paragraph. Please reconsider.
3_ “to improve the safety of highway network through strategy and data-driven methods”. Please provide citations to strenghten your views.
4_ The objective statement is poor (“This paper studies the road paved with steel slag in Yunnan”). What is the gap the authors wish to bridge? What are the scientific gaps and the expected prospects.
5_ The reviewer does not understand the term “depth of construction” in the middle of the slab shown in figure 3. Do you mean texture depth from the “sand patch” test method shown in Figure 4b? Please explain and be more strict with the terminology used. In addition, a brief description of the texture reference testing with its pros and cons is missing. You may consult and cite other similar and reference studies in the research of pavement texture (e.g., http://dx.doi.org/10.1201/9781351063265-86).
6_ Add units in every table wherever applicable.
7_ In section 4.2 about the friction decay rates, did the authors perform any kind of statistical testing to assess the significance in the observed differences? Is the difference in rates from 3.5-6% indeed critical? And with what impact on pavement evaluation and decision-making. This is rather unclear. Please elaborate.
8_ Again, BPN values in table 12 appear to be nearly identical. Please provide justification.
9_ The reviewer would like to see some missing discussion points about the liaison between BPN measures and textural components. Which component is more determinant? Please consult discussion points from other relevant studies, e.g., https://doi.org/10.3390/vehicles2010004.
10_ As an overall comment, the authors should comment on the sustainability gain from the use of steel slag for surface courses. Besides, this was the main triggering point at the beginning of the paper for the reutilization of steel slag. Please reconsider.
My recommendation for this paper is that the authors’ experimental work has indeed some merit, but more effort is needed to improve the scientific nature and structure/presentation of this paper so that it can be reconsidered.
Author Response
Please see the attachment.
I have uploaded the revised PDF file in the attachment. If there is anything in the responses that you do not understand, you can view it in the latest PDF.

Reviewer 3 Report
Hi Authors
Thanks for submission to MDPI.
Title
Analysis of Surface Texture of a Steel Slag–Asphalt Mixture Using Three-Dimensional Scanning Technology.......Kindly revisit
1. Use Asphalt mixtures modified with steel slag in place of Steel Slag–Asphalt Mixture
2. use 3D in place of Three-Dimensional
Abstract
SMA - 13 - Pls use full name
No 3D result finding discussed in abstract? why
Abstract described only about lab tests ? why , The title is inappropriate if lab. work will be described or update it to bring novelty more clearer?
What is the rational of using skid resistance test?
What is grain depth? Pls expand quantitatively?
Keywords: Pls use few from the special issue list Materials | Special Issue : Industrial Solid Wastes for Construction and Building Materials (mdpi.com)
Introduction
Line numbers missing as per MDPI format
the total amount of steel slag accumulated in China is about 1 billion ton --- Pls provide some other developed countries statistics also.
Europe is not country pls correct
Steel slag causes severe pollution to the environment like ---------?/? Can you enter a table about the information?
One of the main reasons that steel slag can be highly utilised in developed countries is that nearly 50% of steel slag is used in road construction-----Reference ????
Steel slag has a high strength, and its mechanical properties are much better than rolled crushed stone-----?/?Can you list few in Table???
Due to the high alkalinity of steel slag (Pls provide typical range and reference), it has strong adhesion to asphalt (Pls provide typical range and reference???)
What is BPN?
Reference pls check as MDPI format?
At the end of 2015, China built the first field full-scale ring road and used 19 kinds of asphalt pavement structures to study the long-term service performance and evolution law of asphalt pavement structures and materials-----Can you list few in table ? Can you compare it with some international findings?
Yunnan - How International reader know about that is city or province?
Kindly enter at the end of introduction a research gap, research motivation, and clearer research objectives.
Table 1 what do you mean by experimental project?
Technical requirements (Table 1) Reference pls???
What are physical and what are mechanical properties in Table 1 Pls separate
meet the requirements of “Technical Specification for Construction of Highway Asphalt Pavement” ??/? Is it some Chinese specifications? Pls provide reference. Remember to make article interesting for international researchers try to use international equivalent nomenclature
f-Cao pls use full name than abbreviation?
f-Cao test carried out by any International or National standard if so pls specify ASTM/ AASHTO/ EN/ BS/CN reference.
Table 1 ASTM/ AASHTO/ EN/ BS/CN reference of tests missing?
Table 2 why results are not shown in result section of paper?
Table 3 ASTM/ AASHTO/ EN/ BS/CN reference of tests missing?
Table3 why tests on lime stone are different from Table 1? Pls add a column in Table 1 and provide info about limestone also?
What is ore powder? Table 4 ASTM/ AASHTO/ EN/ BS/CN reference of tests missing?
Table 5 ASTM/ AASHTO/ EN/ BS/CN reference of tests missing?
Tables 1,2,3,4, 5 what is the source of technical requirement range?
Figure 2 pls improve the quality and label the main components?
Pls tell something about software processing of the ZGScan also?
Table 6 supplier provided information can be referred also?
Table 7 can you draw a gradation curve?
2.5 Experimental methods in place of method typo
BPN pls use full name first? ASTM/ AASHTO/ EN/ BS/CN reference of tests missing?
What is rutting test ASTM/ AASHTO/ EN/ BS/CN reference of tests missing?
Figure 5 not clear? Pls label also? It seems that two asphalt colors why?
Figure 6 can you provide a physical photograph of test section?
How Figure 8 was achieved not clear?
What is rationale of Figure 7 not clear?
Figure 9 the input and out put parameters of the software are not clear?
3.1. High-Temperature Stability tests, Low Temperature stability tests ??? You have not mentioned in materials and methods about that test standard?
Table 8 units missing
How expansion was measured?
We conducted the structural depth test, water penetration test, BPN pendulum test and three-dimensional scanning for one???/ Where are the results? Why different nomenclature?
Equations 1 -7 pls provide reference
Table 10 no rationale pls reconsider?
Table 11 lacks repeatability?
For different groups mentioned and MP-1 to MP-7 pls provide reference gradation?
For mixes? What was optimum binder content? What was air voids? What was specific gravity?
Have you performed indirect tensile strength test?
Can you provide performance grade of the asphalt used?
How Figure 13 was achieved in black and white from colored?
Why Figure 14 is not colored and labelled from the software?
Scales of Figure 13 and 14 are missing? Similarly red color described is missing.
Figure 15 how achieved not clear? Figure 15 sclae missing? Why the output of software was not taken and redrawing was carried out? There seems to be some gap?
Figure 16 at what location the section was cut?
Figure 17 not clear what is the rational?
Conclusions
Pls use bullets and should be extracted from the objectives?
References
Pls use few more from the MDPI
Author Response

(The authors gave the same response as above.)

Round 2
Reviewer 2 Report
The paper was partially improved. Some of my initial points were not properly addressed. My suggestion remains the same.
1_ Lines 107-114 describe the methodology, not the objective. Please revise necessarily.
2_ Again, are the observed differences in the decay rate statistically significant? Please comment.
3_ Initial point 9. I meant which textural component appears to be critical for BPN, micro- or macro-texture. Please revise and provide discussion points.
4_ What did the authors mean by this comment at the end of the response letter. Please be accurate. “At the end, I will modify the INTRODUCTION section by adding a part of the content, specifically modified in Word and PDF.”
5_ Simply adding new references without substantially enriching the main text is not a good practice. Please elaborate further in all cases were new references were used or will be used in the second re-review phase.
Reviewer 3 Report
Hi Authors
Thanks for submission of revised version.
Title: Analysis of Surface Texture of a Asphalt Mixtures Modified 2 with Steel Slag Using 3D Scanning Technology-------- Pls revisit as below
Analysis of Asphalt Mixtures Modified with Steel Slag Surface Texture Using 3D Scanning Technology
Abstract: in the early stages of the test section------- what are these stages? Pls clarify?
Two mixes (Pls name the mixes)
Marshall awellability??? Typo
Tests on trial sections were carried 15 out to assess mean texture depth (MTD) test, British pendulum number (BPN) test were performed 16 for comparison with in-lab tests,------- Wrong sentence
the water immersion Marshall stability 21 and dynamic stability were significantly improved??/ How much please quantify???
with little difference in texture depth, while the former formed more peak tips than the latter-----Pls quantify?
and which can be used as parameters to describe steel slag asphalt pavements. ----- Sentence not clear?
Keywords
height parameters; morphological parameters; ?? what are these parameters and where you have used these?
Introduction
Line 38 the area has become------? Not clear
Line 40 the 39 utilization rate of steel slag can reach 90%.------ In roads???
Line 46 bearing capacity ------ Pls replace with term stiffness??
Line 77 Lal [29-31] -----Pls check citation style
Line 78 Serigos, Andre, and Prozzi [32] ------ Pls check citation style
Line 96 t the end 96 of 2015, China built the first field full-scale ring road and used 19 kinds of asphalt pave- 97 ment structures to study the long-term service performance and evolution law of asphalt 98 Materials 2023, 16, x FOR PEER REVIEW 3 of 25 pavement structures and materials [45], pavement structure includes strong base thin sur- 99 face type semi-rigid base structure, rigid base structure and stress absorbing layer struc- 100 ture, common forms of asphalt pavement structure, asphalt concrete and semi-rigid base 101 combination form, asphalt concrete and semi-rigid base combination form, and full-thick- 102 ness asphalt pavement structure, etc. Loading of the pavement by means of accelerated 103 loading.-----Very long sentence pls divide.
Line 109 are 109 used; from the microscopic point of view, the initial textures of the two pavements are 110 collected using the 3D scanning technique to observe the differences between the two----Pls use past tense were in place of are
Line 139 In order to solve the problem of expansion of steel slag, it should be stored outdoors for several 140 months or more than half a year to promote its expansion and damage------ Pls elaborate under which specification??/ What value will be desired? What was initial value?
Table 3 what is this Robustness (%) index? ? Kindly explain
Line 157 Figure 2a.-----Typo
Line 174 (Figure 8e).------Typo
Line 176 (Figure 8f).------Typo
Line 186 former and the 186 latter is about 0.2 --- Reference
Figures pls bring as per MDPI format
Line 191 what is type I and type II
Authors have not mentioned in materials about asphalt mixtures their mix properties preparation etc? which is their key material as per title
Line 216 No caption of Figure
Line 198 Laboratory experiment??? Pls give headings of experiments like wheel tracker test----- Currently they are mixed and confusing to understand?
Figure numbers are wrongly cited Pls correct in the text
Line 240 Field test or Field pavement test---- Pls revisit?
Line 243 I mixeture -----Typo
Pls give separate heading for BPN and MTT and 3D Scanning under field pavement tests then brief about them in current form they are mixed up and confusing?
Line 289 2.5.3. The Performance Evaluation of Asphalt Mixture ----- Pls bring under laboratory experiment section 2.5.1
Pls provide results in the same sequence as tests were presented in methodology i.e. Laboratory tests,-----------1. 2. 3.-----Field Pavement Tests ---------1. 2. 3.
Line 305 to 314 what is the use of these???
What is the difference of two types of asphalt mixes like asphalt content? air voids? stability ? flow? resilient modulus? etc not clear
Figure 11 why results of three tests??
Figure 12 why results of two tests?
How steel slag modification was carried out ? what was the percentage of steel slag used in asphalt mixtures?
How basalt mixtures were prepared?
No results of field pavement tests shown? why
Line 341 Discussion
Pls provide discussion in the same sequence as tests were presented in methodology i.e. Laboratory tests,-----------1. 2. 3.-----Field Pavement Tests ---------1. 2. 3.
Figure 20 what is the significance of the correlation? what benefit can be achieved from it?
Any logic to correlate 3d texture scanning with skid resistance? You have described lot of other laboratory and field tests and other possible correlations with some rational?
Conclusions must be derived from the objectives?
Pls revisit and try to give conclusions in bullets or 1,2,3 Sr.
Pls add a paragraph at the end about practical application of this research.
